# Uncovering the wider impact of COVID-19 measures on the lives of children with complex care needs and their families: A mixed-methods study protocol

Janet A. Curran[1,2]*, Jennifer Lane[1,2], Holly McCulloch[1], Lisa Keeping-Burke[3], Catie Johnson[1], Helen Wong[2], Christine Cassidy[2], Jessie-Lee McIsaac[4], De-Lawrence Lamptey[4], Julie Clegg[1], Neil Forbes[3], Sydney Breneol[2], Jordan Sheriko[1], Shauna Best[1], Stacy Burgess[1], Doug Sinclair[1], Annette Elliot Rose[1], Mary-Ann Standing[5], Mari Somerville[1], Sarah King[1], Shelley Doucet[3], Heather Flieger[6], Margie Lamb[7], Jeanna Parsons Leigh[2], Dana Stewart[8]

1 IWK Health Centre, Halifax, NS, Canada, 2 Department of Health, Dalhousie University, Halifax, NS, Canada, 3 Department of Nursing & Health Sciences, University of New Brunswick, Fredericton, NB, Canada, 4 Department of Child and Youth Study, Mount Saint Vincent University, Halifax, NS, Canada, 5 Centre for Health and Community Research, University of Prince Edward Island, Charlottetown, PEI, Canada, 6 Queen Elizabeth Hospital, Charlottetown, PEI, Canada, 7 Nova Scotia Health Authority, Halifax, NS, Canada, 8 Parent Partner, Canada

* jacurran@dal.ca

**Data Availability Statement:** No datasets were generated or analysed during the current study. All

## Abstract

Existing barriers to care were exacerbated by the development and implementation of necessary public health restrictions during the COVID-19 pandemic. Children with complex care needs and their families represent a small portion of the paediatric population, and yet they require disproportionately high access to services. Little is known about the impact of COVID-19 public health measures on this population. This study will generate evidence to uncover the wider impact of COVID-19 measures on the lives of children with complex care needs and their families in relation to policy and service changes. This multi-site sequential mixed methods study will take place across the Canadian Maritime provinces and use an integrated knowledge translation approach. There are two phases to this study: 1) map COVID-19 public health restrictions and service changes impacting children with complex care needs by conducting an environmental scan of public health restrictions and service changes between March 2020 and March 2022 and interviewing key informants involved in the development or implementation of restrictions and service changes, and 2) explore how children with complex care needs and their families experienced public health restrictions and service changes to understand how their health and well-being were impacted.

## Introduction

This study addresses a gap in the literature pertaining to children with complex care needs and their families, and how they were impacted by COVID-19 public health restrictions and

relevant data from this study will be made available upon study completion.

**Funding:** This work is supported by Canadian Institutes of Health Research, Operating Grant: Addressing the Wider Health Impacts of COVID-19. The funders had no role in study design, data collection and analysis, decision to publish, or preparation of the manuscript.

**Competing interests:** The authors have declared that no competing interests exist.

service changes. Since the emergence of COVID-19 in late 2019, rapid implementation of community-based public health measures have aimed to reduce transmission of the virus [1]. In Canada, provincial public health authorities were responsible for setting policy recommendations and restrictions [2]. While public health restrictions were necessary to slow community transmission, reduce deaths, and protect the health care system, they have negatively affected various Canadian communities, particularly those that already experience barriers to care [3].

Public health measures related to COVID-19 have directly impacted how children and their families access care, interact, socialize, and participate in everyday society. Emerging literature from the Canadian province of Ontario reveals that COVID-19 related public health measures led to unintended adverse outcomes for children and their families related to mental health, food security, physical activity, access to health services, and community supports [4]. Additional evidence suggests these impacts may have been exacerbated for vulnerable pediatric populations, including children with complex needs [5] a population with "multidimensional health and social care needs, in the presence of a recognized medical condition or where there is no unifying diagnosis" [6]. Further, there is a conspicuous gap in the literature related to how the sociostructural determinants of health, including employment, education, public infrastructure, criminal justice, child welfare, food systems, and health care [7–9], were experienced by these children and their families during COVID-19. As such, the experiences of children with complex care needs who face more barriers to care due to sociostructural determinants of health—and their families—are effectively erased.

While the population of children with complex care needs is relatively small in numbers, they require extensive health, educational, and social resources [10–13]. The pre-pandemic disjuncture between health and social support systems and family-identified needs means caregivers assume the burden of intensive caregiving, care coordination, home-based medical care, and associated expenses [14–16]. The pandemic likely widened the gap in care for these families with changes to available health resources such as the reduction or cancellation of non-urgent services, ambulatory care, and in-home supports. The influence of sociostructural determinants of health on those whose underservicing in health care pre-existed COVID-19 is thus clear; social inequities perpetuating health disparities were exacerbated during the pandemic for many who were already marginalized within dominant healthcare structures [3]. Moreover, the impacts of switching to virtual learning and the availability of resources typically provided to children with complex needs in the classroom effectively removed a source of respite for many families [17, 18].

The Canadian Maritimes present a unique context to explore the impacts of COVID-19 related public health measures for children and families with complex care needs. The three Maritime provinces were required to work together to coordinate and communicate public health measures that had cross-provincial implications. Further, there is only one pediatric tertiary care facility (IWK Health Centre) in the Maritimes mandated to provide specialty care services for children and families. Although the rapid response in the Maritimes has been recognized globally for successfully reducing the burden of COVID-19 [19], the wider impact of these public health measures on children with complex care needs and their families remains unclear.

Our previous work examining children with complex needs and their families highlighted several potential quality and safety challenges that must be explored in the context of the COVID-19 pandemic. Of importance, we identified several policy and service gaps related to care navigation across hospital and community settings for this diverse population in the Maritimes [16]. Many families also described the burden they experienced advocating for access to formal and informal supports to meet their needs. The prevailing gaps in our health care and

social support structures highlight the potential cascading impacts of COVID-19 restrictions on families.

## Methods and analysis

### Aim and objectives

The overall aim of this research is to understand how COVID-19 public health restrictions and service changes impacted the health and well-being of children with complex care needs and their families in the Canadian Maritime provinces: Nova Scotia (NS), New Brunswick (NB), and Prince Edward Island (PE).

We will highlight innovative practices and areas for improvement and translate this understanding into recommendations for policy reform. We will achieve this overall aim through the following objectives (activities for each are described in Table 1):

1. Map COVID public health restrictions and service changes impacting children with complex care needs.

2. Explore how children with complex care needs and their families experienced public health restrictions and service changes to understand how their health and well-being were impacted.

### Setting

This study will take place at three sites across the Canadian Maritime provinces of NS, PEI, and NB.

### Design

We will use a sequential mixed methods study design [20] and an integrated knowledge translation (iKT) approach [21] to explore the experiences of children with complex care needs and their families in relation to policy and service changes in the Maritimes during the COVID-19 pandemic. Participants who wish to discuss impacts of the pandemic that they are still experiencing will be supported in doing so. This study will be carried out in two phases. The first phase will include 1) identifying COVID-19 public health restrictions and service changes between March 2020 and March 2022, 2) describing common public health restrictions and service change implementation strategies, and 3) identifying perceived barriers and enablers to implementation of public health restrictions and service changes. The second phase will include: 1) describing the family-reported impact of COVID-19 public health restrictions and service changes, 2) identifying perceived barriers and enablers to navigation, and 3) describing policy and service gaps identified by children and families during the COVID-19 pandemic.

**Table 1. Activities by objective.**

|  | Activities |
|---|---|
| **Objective 1** | • Identify COVID-19 public health restrictions and service changes between March 2020 and March 2022<br>• Describe common public health restrictions and service change implementation strategies<br>• Identify perceived barriers and enablers to implementation of public health restrictions and service changes |
| **Objective 2** | • Describe the family-reported impact of COVID-19 public health restrictions and service changes<br>• Identify perceived barriers and enablers to navigating COVID-19 public health restrictions and service changes<br>• Describe policy and service gaps identified by children and families during the COVID-19 pandemic |

Meetings involving the full team or parts thereof (depending on which aspect of the research process) will take place regularly with appropriate stakeholders, including youth, parents/guardians/primary caregivers, policy makers, service providers, and other knowledge users.

A sex- and gender-based analysis (SGBA+) will be integrated into the research design to further understandings of sociostructural determinants of health influencing the experiences of children with complex care needs and their families during the COVID-19 pandemic. Paying attention to differences within a sample population during the recruitment phase will do more than allow us to describe our sample; points of comparison will be created and then used later when data are being analyzed to enhance qualitative trustworthiness [22, 23]. The SGBA + will be integrated into research methods by developing a REDCap sampling instrument and collecting participant sociodemographic data pertaining to employer and professional role (key informants only), age, and across PROGRESS+ variables (sex, gender, race, ethnicity, socioeconomic status, (dis)ability, and citizenship). Doing so will inform purposive sampling, support an investigation of health inequities by engaging broader systems of power, and exploration of the impacts of sociostructural factors within the studied context [3, 24, 25].

The sampling instrument will also provide insight into what considerations are required to cultivate psychologically safe environments for conducting culturally appropriate, individualized interviews, and promotes the application of the principles of equity, diversity, and inclusion in our research design. As such, our SGBA+ will promote a complicated exploration of sociostructural determinants of health within the studied context perpetuating gaps in health care and social support structures, impacting children with complex care needs, their families, and the ability of individuals working with them.

## Phase one

**Map COVID-19 public health policy and/or service changes.** We will conduct an environmental scan to identify public health directives, regulations, restrictions and/or service changes implemented in the Maritimes during the COVID-19 pandemic relevant to children with complex care needs and their families. We will then conduct interviews with key informants to further understand the development and implementation of identified policy and service changes.

*Environmental scan.* The project team will work in conjunction with an information specialist at the Maritime SPOR SUPPORT Unit (MSSU) to develop a systematic search of government websites across the three Maritime provinces. A search strategy will first be created at the lead site (IWK, NS) and modified to the context of the collaborating sites. Searches will be followed with secondary navigation aids, including unstructured browsing and manual in-site searches as appropriate. Inclusion and exclusion criteria pertain to nature, source, timeframe, and accessibility of policy changes related to COVID-19 (Table 2).

We will co-design a REDCap data abstraction form with our stakeholder collaborators and anticipate abstracting data related to the type of policy or service, author, organization, resource link, publication date, target population, time limitations, key stakeholders and their involvement, geographic limitations, and policy dissemination and/or implementation strategy. Data will be sorted into categories codesigned by parent partners. Categories will be suggested by the research team, and then refined with parent partners via a consensus meeting, to ensure nuances of the data are captured effectively within this categorization process. The abstraction form will be pilot tested with sources from each province prior to data collection. The research assistant (RA) hired at each site will complete data abstraction for their province and a second reviewer will undergo verification. Discrepancies will be resolved through consensus.

**Table 2. Eligibility criteria by phase.**

| | | Inclusion Criteria | Exclusion Criteria |
|---|---|---|---|
| Phase 1 | Environmental Scan | Documents and webpages published by government, health authorities, or not-for-profit organizations<br>Resources related to COVID-19 policy and service changes AND relevant to children with complex care needs and their families<br>Published between March 2020 and March 2022<br>Public Accessibility<br>English or French language | Directives and/or service changes that were not implemented in response to COVID-19<br>Policies published before March 2020 or after March 2022<br>Language other than English or French |
| | Key Informant Interviews | Involved in designing or implementing COVID-19 public health measures/mandates in the Maritimes<br>English-speaking | |
| Phase 2 | Interviews with Children/Youth and/ or Primary Caregivers/ Legal Guardian | Children/Youth (11–18 years of age) who identify as having complex care needs and/or primary caregivers/legal guardians of children/youth with complex care needs<br>Residing in the Maritimes between March 2020 and March 2022<br>English-speaking | |

Summary tables will be developed to allow for inter-provincial comparisons as will a visual timeline representation of sentinel changes over the study period.

*Key informant interviews.* A draft list of potential key informants from different service sectors in each of the three provinces will be generated in collaboration our decision-maker partners. We will contact potential informants through publicly available email addresses. Demographic data of the informants will be collected using the aforementioned REDCap sampling instrument developed for the purpose of integrating a SGBA+ into our research methods. The instrument will be used to collect sociodemographic data from participants after initial contact has been made, once consent has been obtained, and before the interview. Interviews will aim to explore the categories used for the environmental scan. Key informants will be mapped across these categories according to their role to promote a broad exploration of findings from the environmental scan and gain insight on the decision-making and implications on service delivery in relation to COVID-19 public health policy and service changes. Snowball sampling may help to identify additional key informants and research participants for phase two interviews.

All groups will receive respectful research considerations. This research will be conducted on the unceded and ancestral land of the Mi'kmaq, Peskotomuhkati, Wolastoqiyik Peoples, and our research team recognizes that this comes with responsibilities to the land and each other because we are all Treaty People. While we will not be directly engaging Indigenous communities or organizations, Indigenous persons may become involved in the study, we will take steps to support information governance and data sovereignty with First Nations participants.

Key informant interviews will aim to be under 30 minutes in length, excluding the time it takes to obtain consent and collect sociodemographic data. A semi-structured interview guide, informed by findings from the environmental scan and the Theoretical Domains Framework (TDF) will be developed in consultation with our parent and decision-making partners. The TDF is a synthesis of 33 theories of behaviour change and offers a theoretical lens to view cognitive, affective, social, and environmental influences on implementation behaviour [26]. Interview questions will aim to examine participants' perceptions of barriers and enablers to COVID-19 related policy and service development and implementation [26]. Interviews will take place virtually using Zoom videoconferencing technology. All interviews will be audio recorded and transcribed verbatim.

*Eligibility*. Inclusion criteria are related to role and language spoken. Eligibility criteria for key informant interview participants recruited from multiple government/non-for-profit sectors/programs across the Maritimes include involvement in designing or implementing COVID-19 public health measures/mandates in the Maritimes and being able to speak English (Table 2).

*Participants and sample size*. We aim to recruit 5–12 key informants from each Maritime province for a total of 15–36 participants in the three provinces combined.

*Recruitment and consent*. Potential informants will be contacted through email with a letter of invitation to participate in an interview.

Following the initial contact with the potential participant, a research team member will ask if they can arrange a time to meet with potential participants, briefly introduce the study purpose, obtain verbal consent, and collect sociodemographic data before conducting the interview.

### Phase two

**Impacts on children with complex care needs and their families.** Informed by findings from phase one, phase two will involve qualitative semi-structured interviews with youth living with complex care needs and/or their caregivers/legal guardians. This work will be guided by interpretive phenomenology [27–32], to capture the lived experiences of families with complex care needs in relation to COVID-19 public health policy and service changes.

The sampling instrument used to collect sociodemographic data from key informants will be adapted and serve the same purpose for the child/youth and primary caregiver/legal guardian subgroups, thus extending the integration of a SGBA+ into phase two and promoting our aim to cultivate psychologically safe environments for conducting culturally appropriate, individualized interviews with caregivers/guardians of child participants. Our approach will be guided by established literature outlining inclusive processes to engage children with disabilities in research [33–35]. For example, the sampling instrument will ask participants if there is anything that would help support their participation in this study and will provide examples, such as assistive or adaptive technology. Further, to mitigate the researcher-participant power dynamic, youth and caregiver/guardian interviews will be co-facilitated by a research team member and a parent partner [33, 35]. Guardians need to be present when youth assent is being obtained, but their presence during the interview is not necessary. As with key informants, all groups will receive respectful research considerations; however, community-identified engagement strategies will be used if First Nations families who have children with complex care needs become involved in the study.

Interviews will last approximately 30 minutes. The development of phase two interview guides (one for parents/guardians/primary caregivers and another for youth) will be informed by phase one findings; questions will be developed in relation to the same categories from the environmental scan. Probes will be used to understand families' experiences with COVID-19 specific health, social and educational policy and service changes identified in phase one. Interviews will take place virtually using Zoom videoconferencing technology, using the same process as the key informant interviews. Interviews will be audio recorded and transcribed verbatim.

*Eligibility*. Inclusion criteria are related to age, identity, role, residence, and language spoken. Eligibility criteria for parents and youth interview participants include being a child/youth between the ages of 11–18 years old who identify as having complex care needs or being a caregiver/legal guardian to one, residing in the Maritimes between March 2020 and March 2022, and being English speaking (Table 2).

*Participants and sample size.* We will aim to recruit up to 12–16 participants from each of the Maritime provinces for a total of 36–48 participants from the three provinces, combined. Sampling methods for phase two are the same as those used in phase one [36] and will be enhanced by engaging our parent partners and MSSU consultants to guide our recruitment strategies.

*Recruitment and consent.* We will recruit children/youth and parents/guardians/caregivers in a variety of ways. Recruitment ads will be created (posters, Facebook, and Twitter) and reflect diverse families, health conditions, and ages of children/youth. IWK care team managers will be asked to identify potential families that meet the parameters of inclusion. The research team will prepare mail-outs of a letter of information to families and provide the mail-outs to the care teams and share ads through their own networks via email and social media. Managers will be asked if their department can address and send letters of invitation to those who meet the inclusion criteria. In the letter of information, families will be invited to contact the RA of this study at their site by email or phone if they wish to participate or learn more about the study.

We will share posters and social media ads with the research team, through social media, and the networks of our collaborating partners (e.g. MSSU, Family Leadership Council, Youth Advisory Council). On digital flyers, eligible individuals will be asked to contact the research coordinator via phone or email for more information. In so far as the IWK is a tertiary centre accessed by children/youth with complex needs and their families from across the Maritime provinces, research participants will be recruited within the three Maritime provinces using these methods. Key informant snowball sampling will facilitate recruitment of research participants within each province as for NB and PEI sites, patient registries at provincial health authorities will be accessed with appropriate permission and ethical considerations.

Following the initial contact by the potential participant, a member of the research team will collect contact information and arrange a Zoom meeting with participants willing to take part in the study at a time that is mutually convenient. Meetings will entail a brief introduction to the purpose of the study, review of environmental scan findings, obtaining informed consent verbally, completing the same recruitment questionnaire as key informants in phase 1, and conducting an interview, for a total of approximately one hour.

## Analyses

Thematic analysis will be used to analyze environmental scan data. Directives in each category will be reduced into 1–2 themes and used to develop an interview question with prompts, which may include specific examples from the environmental scan. Those that cannot be reduced will be used to create a Delphi survey that will be sent to the research team. Members of the team will be asked to vote and choose three directives of greatest significance for each category. The top directives will then be used to develop interview questions for that category and added to the interview guide. If the first Delphi survey does not produce a clear top three directives, then a second Delphi survey will be circulated to the same team with only the top 5 directives from the first survey.

Qualitative (interview) data from phase one will be analyzed using a deductive and inductive approach [37]. The Ten Domains of Health (TDH) [38] will be used to guide directed content analysis [39] to further contextualize the environmental scan data. The codes will be the TDH themselves: basic needs; inclusive education; child social integration; current child health-related quality of life; long-term child and self-sufficiency; family social integration; community system supports; heath care system supports; high-quality patient-centered medical home; and family-centered care. The conceptual boundaries of each code will be defined

and a framework for understanding the lived experiences of these children and their families will thus be developed. TDH codes will be used to code all qualitative (interview) data, from both phases. Two reviewers will independently code each transcript using Quirkos; discrepancies in coding will be recorded and resolved through consensus by a third reviewer.

The TDH coding framework will guide an interpretive phenomenological analysis [27–31] of phase two interview data. Data will be analyzed with our research team, including parent partners, to understand participants' lived experiences in relation to COVID-19 public health policy and service changes. Member checking will be used to explore the credibility of findings [40].

We will triangulate data by developing a matrix to compare and contrast key findings from phase one and two. We will use the key Learning Health Systems (LHS) features proposed by Lavis et al. [41] as an organizing structure for our data triangulation matrix: 1) anchored in patient needs, perspectives, and aspirations, 2) driven by timely data and evidence, 3) supported by appropriate decision supports and aligned governance; and 4) enabled with a culture of rapid learning and improvement. This approach to data triangulation will serve as a guide to understand the barriers and enablers for policy and service delivery to support children with complex needs and their families in the Maritimes during unprecedented times such as the COVID-19 pandemic.

## Knowledge translation

We will host a half day consensus meeting, which will include researchers, youth, parents, clinicians, and decision-makers from all three Maritime provinces. During this meeting, we will review findings and achieve the following:

1. Co-develop a list of innovative and promising practices and a list of policy and practice recommendations stemming from our data triangulation;

2. Develop an action plan to disseminate findings and implement policy and practice recommendations.

## Ethics and dissemination

Ethics approval for this study has been obtained from Research Ethics Boards at the IWK Health Centre, the University of New Brunswick, Horizon Health Network, and Health PEI. Informed consent will be obtained from all participants prior to their involvement in the study. Results from the research will highlight innovative practices and areas for improvement and translate this understanding into recommendations for policy reform.

## Discussion

Strengths of this study will include the involvement of a multi-disciplinary team, including parents, clinicians, and researchers, from across all three provincial sites which adds strength to this work through the variety of perspectives. Stakeholder involvement, including parent partners, will enhance processes related to recruitment, data collection and data analysis, and identification of significant findings. Parent partners will be at the centre of this research and will provide their perspectives and lived experiences throughout the duration of the study. Recommendations will be created in collaboration with parent partners at the end of the study, to ensure meaningful outcomes from this work. Additionally, a sex- and gender-based analysis plus (SGBA+) will be integrated into qualitative methods to enhance qualitative trustworthiness and further understandings of sociostructural factors influencing the experiences of

children with complex care needs and their families in relation to public health restrictions and service changes between March 2020 and March 2022.

Limitations include the potential for recall bias in relation to length of time between interviews and the period of change being investigated. To mitigate this limitation, findings from an environmental scan of public health measures and restrictions between March 2020 and March 2022 will be shared with interviewees to clarify the ongoing significance of pandemic restrictions and their impacts on various populations to which children with complex care needs and their families belong. An additional limitation includes the environmental scan being limited to only provincial government news releases to identify public health policy changes, this may exclude findings that were only released in health services announcements.

## Acknowledgments

We would like to thank the MSSU librarian scientist with their help designing the search strategy.

## Author Contributions

**Conceptualization:** Janet A. Curran.

**Funding acquisition:** Janet A. Curran.

**Methodology:** Holly McCulloch, Lisa Keeping-Burke, Christine Cassidy.

**Project administration:** Jennifer Lane.

**Supervision:** Janet A. Curran, Jennifer Lane, Holly McCulloch.

**Writing – original draft:** Jennifer Lane, Holly McCulloch.

**Writing – review & editing:** Janet A. Curran, Holly McCulloch, Lisa Keeping-Burke, Catie Johnson, Helen Wong, Christine Cassidy, Jessie-Lee McIsaac, De-Lawrence Lamptey, Julie Clegg, Neil Forbes, Sydney Breneol, Jordan Sheriko, Shauna Best, Stacy Burgess, Doug Sinclair, Annette Elliot Rose, Mary-Ann Standing, Mari Somerville, Sarah King, Shelley Doucet, Heather Flieger, Margie Lamb, Jeanna Parsons Leigh, Dana Stewart.

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
