## [Decision Letter · Decision Letter 0]

16 Oct 2023

PONE-D-23-22003Uncovering the Wider Impact of COVID-19 Measures on the Lives of Children with Complex Care Needs and their Families: a mixed-methods study protocolPLOS ONE

Dear Dr. Curran,

Thank you for submitting your manuscript to PLOS ONE. After careful consideration, we feel that it has merit but does not fully meet PLOS ONE’s publication criteria as it currently stands. Therefore, we invite you to submit a revised version of the manuscript that addresses the points raised during the review process.

We look forward to receiving your revised manuscript.

Kind regards,

Jerome Nyhalah Dinga, PhD

Academic Editor

PLOS ONE

“This work is supported by Canadian Institutes of Health Research, Operating Grant: Addressing the Wider Health Impacts of COVID-19. Grant #473640 Dr. Janet Curran is the Nominated Principal Applicant”

3. We note that you have referenced (Lane J. Operationalizing Intersectionality Theory in Research. Dalhous Univ. in preparation) which has currently not yet been accepted for publication. Please remove this from your References and amend this to state in the body of your manuscript: (ie “Bewick et al. [Unpublished]”) as detailed online in our guide for authors

Reviewers' comments:

Reviewer's Responses to Questions

**Comments to the Author**

1. Does the manuscript provide a valid rationale for the proposed study, with clearly identified and justified research questions?

Reviewer #1: Yes

2. Is the protocol technically sound and planned in a manner that will lead to a meaningful outcome and allow testing the stated hypotheses?

Reviewer #1: Partly

3. Is the methodology feasible and described in sufficient detail to allow the work to be replicable?

Reviewer #1: Yes

4. Have the authors described where all data underlying the findings will be made available when the study is complete?

Reviewer #1: Yes

5. Is the manuscript presented in an intelligible fashion and written in standard English?

Reviewer #1: Yes

6. Review Comments to the Author

You may also provide optional suggestions and comments to authors that they might find helpful in planning their study.

Reviewer #1: This is an interesting protocol to research the effects of the COVID 19 measures on the care of children with complex needs.

It is not clear however, what is meant by complex needs and from the way the protocol is presented there is no specific gradient regarding the severity of each case. From what it is written, the interview in Phase 2 will be with the child, so cases of complex needs that impede communication are hereby excluded.

Recall bias is an important limitation. Perhaps it would be helpful to carefully map the current state of care of these children (post pandemic) with information collected during the interview as well as from the services (eligibility for benefits, visits to the IWK health care unit etc) to facilitate comparison with the pandemic period.

7. PLOS authors have the option to publish the peer review history of their article (what does this mean?). If published, this will include your full peer review and any attached files.

Reviewer #1: No

---

## [Author Response · Author response to Decision Letter 0]

14 Dec 2023

Dear editor,

Thank you for providing reviewer feedback. We have included our responses below and have provided the revised manuscript. We believe we have addressed all the comments and that the suggestions have improved the paper.

Thank you. All the files are in accordance with the required naming convention and the manuscript meets the style requirements.

“This work is supported by Canadian Institutes of Health Research, Operating Grant: Addressing the Wider Health Impacts of COVID-19. Grant #473640 Dr. Janet Curran is the Nominated Principal Applicant”

Thank you. We have amended the financial disclosure as advised in both the manuscript and the cover letter.

3. We note that you have referenced (Lane J. Operationalizing Intersectionality Theory in Research. Dalhous Univ. in preparation) which has currently not yet been accepted for publication. Please remove this from your References and amend this to state in the body of your manuscript: (ie “Bewick et al. [Unpublished]”) as detailed online in our guide for authors

Thank you. We have removed all references that have not yet been accepted for publication.

Thank you. We have removed the ethics statement from all sections besides the Methods.

Thank you. A full review of the reference list was performed, and it is complete and correct. Reference 26 was updated from the accepted to the published citation.

Comments to the Author

1. Does the manuscript provide a valid rationale for the proposed study, with clearly identified and justified research questions?

Reviewer #1: Yes

Thank you.

2. Is the protocol technically sound and planned in a manner that will lead to a meaningful outcome and allow testing the stated hypotheses?

Reviewer #1: Partly

Thank you. We have added further clarity to the definition of complex care that we are using for this project to increase the clarity.

3. Is the methodology feasible and described in sufficient detail to allow the work to be replicable?

Reviewer #1: Yes

Thank you.

4. Have the authors described where all data underlying the findings will be made available when the study is complete?

Reviewer #1: Yes

Thank you.

5. Is the manuscript presented in an intelligible fashion and written in standard English?

Reviewer #1: Yes

Thank you.

6. Review Comments to the Author

You may also provide optional suggestions and comments to authors that they might find helpful in planning their study.

Reviewer #1: This is an interesting protocol to research the effects of the COVID 19 measures on the care of children with complex needs.

It is not clear however, what is meant by complex needs and from the way the protocol is presented there is no specific gradient regarding the severity of each case. From what it is written, the interview in Phase 2 will be with the child, so cases of complex needs that impede communication are hereby excluded. 

Thank you. As we state in lines 74-77, complex care needs for this project are defined as multidimensional health and social care needs. The Phase 2 interviews will be conducted with youth and/or caregivers. Therefore, should a child have a case of complex needs that impedes communication, their caregiver will be interviewed. 

Recall bias is an important limitation. Perhaps it would be helpful to carefully map the current state of care of these children (post pandemic) with information collected during the interview as well as from the services (eligibility for benefits, visits to the IWK health care unit etc) to facilitate comparison with the pandemic period.

Thank you. We agree that recall bias is an important limitation to address in this work. This project is looking specifically at the impact during the pandemic, so we do not intend to compare to the post-pandemic period. We have included the environmental scan to have a point of reference to remind both key informants and families of the key changes during the pandemic to clarify their impact.

---

## [Editor Report · Decision Letter 1]

2 Jun 2024

Uncovering the Wider Impact of COVID-19 Measures on the Lives of Children with Complex Care Needs and their Families: a mixed-methods study protocol

PONE-D-23-22003R1

Dear Dr. Curran,

We’re pleased to inform you that your manuscript has been judged scientifically suitable for publication and will be formally accepted for publication once it meets all outstanding technical requirements.

Kind regards,

Jianhong Zhou

Staff Editor

PLOS ONE
---

## [Editor Report · Acceptance letter]

4 Jun 2024

PONE-D-23-22003R1 

PLOS ONE

Dear Dr. Curran, 

I'm pleased to inform you that your manuscript has been deemed suitable for publication in PLOS ONE. Congratulations! Your manuscript is now being handed over to our production team.

Kind regards, 

on behalf of

Dr. Jianhong Zhou 

Staff Editor

PLOS ONE